# PRACTICAL LOSSLESS COMPRESSION WITH LATENT VARIABLES USING BITS BACK CODING

**James Townsend, Thomas Bird & David Barber**
Department of Computer Science
University College London
{james.townsend,thomas.bird,david.barber}@cs.ucl.ac.uk

## ABSTRACT

Deep latent variable models have seen recent success in many data domains. Lossless compression is an application of these models which, despite having the potential to be highly useful, has yet to be implemented in a practical manner. We present 'Bits Back with ANS' (BB-ANS), a scheme to perform lossless compression with latent variable models at a near optimal rate. We demonstrate this scheme by using it to compress the MNIST dataset with a variational auto-encoder model (VAE), achieving compression rates superior to standard methods with only a simple VAE. Given that the scheme is highly amenable to parallelization, we conclude that with a sufficiently high quality generative model this scheme could be used to achieve substantial improvements in compression rate with acceptable running time. We make our implementation available open source at https://github.com/bits-back/bits-back.

## 1 INTRODUCTION

The connections between information theory and machine learning have long been known to be deep, and indeed the two fields are so closely related that they have been described as 'two sides of the same coin' (Mackay, 2003). One particularly elegant connection is the essential equivalence between probabilistic models of data and lossless compression methods. The source coding theorem (Shannon, 1948) can be thought of as the fundamental theorem describing this idea, and Huffman coding (Huffman, 1952), arithmetic coding (Witten et al., 1987) and the more recently developed asymmetric numeral systems (Duda, 2009) are actual algorithms for implementing lossless compression, given some kind of probabilistic model.

The field of machine learning has experienced an explosion of activity in recent years, and we have seen a number of papers looking at applications of modern deep learning methods to lossy compression. Gregor et al. (2016) discusses applications of a deep latent Gaussian model to compression, with an emphasis on lossy compression. Ballé et al. (2017), Theis et al. (2017), Ballé et al. (2018), and Minnen et al. (2018) all implement lossy compression using (variational) auto-encoder style models, and Tschannen et al. (2018) train a model for lossy compression using a GAN-like objective. Applications to lossless compression have been less well covered in recent works. We seek to advance in this direction, and we focus on lossless compression using latent variable models.

The lossless compression algorithms mentioned above do not naturally cater for latent variables. However there is a method, known as 'bits back coding' (Wallace, 1990; Hinton and van Camp, 1993), first introduced as a thought experiment, but later implemented in Frey and Hinton (1996) and Frey (1997), which can be used to extend those algorithms to cope with latent variables.

Although bits back coding has been implemented in restricted cases by Frey (1997), there is no known efficient implementation for modern neural net-based models or larger datasets. There is, in fact, a fundamental incompatibility between bits back and the arithmetic coding scheme with which it has previously been implemented. We resolve this issue, describing a scheme that instead implements bits back using asymmetric numeral systems. We term this new coding scheme 'Bits Back with ANS' (BB-ANS).

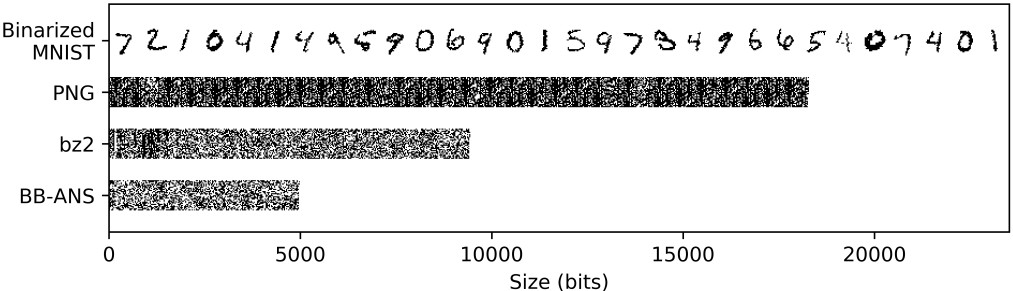

Figure 1: Visual comparison of 30 binarized MNIST images with bitstream outputs from running lossless compression algorithms PNG, bz2 and BB-ANS on the images[1].

Our scheme improves on existing implementations of bits back coding in terms of compression rate and code complexity, allowing for efficient lossless compression of arbitrarily large datasets with deep latent variable models. We demonstrate the efficiency of BB-ANS by losslessly compressing the MNIST dataset with a variational auto-encoder (VAE), a deep latent variable model with continuous latent variables (Kingma and Welling, 2013; Rezende et al., 2014). As far as we are aware, this is the first time bits back coding has been implemented with continuous latent variables.

We find that BB-ANS with a VAE outperforms generic compression algorithms for both binarized and raw MNIST, even with a very simple model architecture. We extrapolate these results to predict that the performance of BB-ANS with larger, state of the art models would be significantly better than generic compression algorithms.

## 2   BITS BACK CODING

In this section we describe bits back coding, a method for lossless compression of data using a latent variable model. Before we describe bits back itself, we briefly discuss methods for encoding a stream of data given a fully observed model, a task sometimes referred to as 'range coding' or 'entropy coding'. We do not go into detail about the algorithms or their implementation, but describe the high level characteristics necessary for understanding bits back.

For brevity, in the following sections we use simply $\log$ to refer to the base 2 logarithm, usually denoted $\log_2$. Message lengths are measured in bits.

### 2.1   COMPRESSING STREAMS WITH ARITHMETIC CODING VS. ASYMMETRIC NUMERAL SYSTEMS

Suppose that someone ('the sender') has a sequence of randomly distributed symbols, $s = (s_1, ..., s_N)$, with each $s_n$ drawn from a finite alphabet $\mathcal{A}_n$, which they would like to communicate to someone else ('the receiver') in as few bits as possible. Suppose that sender and receiver have access to a probabilistic model $p$ for each symbol in the sequence given the previous, and can compute the mass $p(s_n = k \,|\, s_1, \ldots, s_{n-1})$ for each $k \in \mathcal{A}_n, n \in \{1, \ldots, N\}$.

Arithmetic coding (AC) and asymmetric numeral systems (ANS) are algorithms which solve this problem, providing an encoding from the sequence $s$ to a sequence of bits (referred to as the 'message'), and a decoding to recover the original data $s$. Both AC and ANS codes have message length equal to the 'information content' $h(s) \triangleq -\log p(s)$ of the sequence plus a small constant overhead of around 2 bits. By Shannon's Source Coding Theorem, the expected message length can be no shorter than the entropy of the sequence $s$, defined by $H[s] \triangleq \mathbb{E}[h(s)]$, and thus AC and ANS are both close to optimal (Shannon, 1948; Mackay, 2003). For long sequences the small constant overhead is amortized and has a negligible contribution to the compression rate.

---

[1]Code to reproduce this figure is in the git repository, filename `make_fig_1.py`.

Critically for bits back coding, AC and ANS differ in the order in which messages are decoded. In AC the message is FIFO, or queue-like. That is, symbols are decoded in the same order to that in which they were encoded. ANS is LIFO, or stack-like. Symbols are decoded in the opposite order to that in which they were encoded.

Note that the decoder in these algorithms can be thought of as a mapping from i.i.d. bits with $p(b_i = 0) = p(b_i = 1) = \frac{1}{2}$ to a sample from the distribution $p$. Since we get to choose $p$, we can also think of ANS/AC as invertible samplers, mapping from random bits to samples via the decoder and back to the same random bits via the encoder.

For a far more detailed introduction to arithmetic coding, see Witten et al. (1987), for asymmetric numeral systems, see Duda (2009).

## 2.2 BITS BACK CODING

We now give a short description of bits back coding, similar to those that have appeared in previous works. For a more involved derivation see Appendix A. We assume access to a coding scheme such as AC or ANS which can be used to encode and decode symbols according to any distribution. We will return to the question of which is the correct coding scheme to use in Section 2.4.

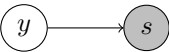

Figure 2: Graphical model with latent variable $y$ and observed variable $s$.

Suppose now a sender wishes to communicate a symbol $s_0$ to a receiver, and that both sender and receiver have access to a generative model with a latent variable, $y$. For now we take $y$ to be discrete, we address continuous latents in Section 2.5.1. Suppose both sender and receiver can compute the forward probabilities $p(y)$ and $p(s \mid y)$, and also have access to an approximate posterior $q(y \mid s)$. Bits back coding allows the sender and receiver to efficiently encode and decode the symbol $s_0$.

We must assume that, as well as the sample $s_0$, the sender has some extra bits to communicate. The sender can *decode* these extra bits to generate a sample $y_0 \sim q(y \mid s_0)$. Then they can encode the symbol $s_0$ according to $p(s \mid y_0)$ and the latent sample according to $p(y)$. The receiver then does the inverse to recover the latent sample and the symbol. The extra bits can also be recovered by the receiver by *encoding* the latent sample according to $q(y \mid s_0)$. We can write down the expected increase in message length (over the extra bits):

$$L(q) = \mathbb{E}_{q(y \mid s_0)}\big[ -\log p(y) - \log p(s_0 \mid y) + \log q(y \mid s_0)\big] \tag{1}$$

$$= -\mathbb{E}_{q(y \mid s_0)} \log \frac{p(s_0, y)}{q(y \mid s_0)}. \tag{2}$$

This quantity is equal to the negative of the evidence lower bound (ELBO), sometimes referred to as the 'free energy' of the model.

A great deal of recent research has focused on inference and learning with approximate posteriors, using the ELBO as an objective function. Because of the above equivalence, methods which maximize the ELBO for a model are implicitly minimizing the message length achievable by bits back coding with that model. Thus we can draw on this plethora of existing methods when learning a model for use with bits back, safe in the knowledge that the objective function they are maximizing is the negative expected message length.

## 2.3 CHAINING BITS BACK CODING

If we wish to encode a *sequence* of data points, we can sample the extra bits for the first data point at random. Then we may use the encoded first data point as the extra information for the second data point, the encoded second data point as the extra information for the third, and so on. This daisy-chain-like scheme was first described by Frey (1997), and was called 'bits-back with feedback'. We refer to it simply as 'chaining'.

As Frey (1997) notes, chaining cannot be implemented directly using AC, because of the order in which data must be decoded. Frey gets around this by implementing what amounts to a stack-like

wrapper around AC, which incurs a cost both in code complexity and, importantly, in compression rate. The cost in compression rate is a result of the fact that AC has to be 'flushed' in between each iteration of bits back, and each flush incurs a cost which is implementation dependent but typically between 2 and 32 bits.

## 2.4 Chaining bits back coding with ANS

The central insight of this work is to notice that the chaining described in the previous section can be implemented straightforwardly with ANS with zero compression rate overhead per iteration. This is because of the fact that ANS is stack-like by nature, which resolves the problems that occur if one tries to implement bits back chaining with AC, which is queue-like. We now describe this novel method, which we refer to as 'Bits Back with ANS' (BB-ANS).

We can visualize the stack-like state of an ANS coder as

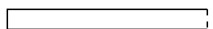

where the dashed line symbolizes the encoding/decoding end or 'top' of the stack. When we encode a symbol $s$ onto the stack we effectively add it to the end, resulting in a 'longer' state

$$-\log p(s)$$

and when we decode (or equivalently, sample) a symbol $t$ from the stack we remove it from the same end, resulting in a 'shorter' state, plus the symbol that we decoded.

$$-\log p(t)$$

$, \quad t$

Table 1 shows the states of the sender as they encode a sample, using our bits back with ANS algorithm, starting with some 'extra information' as well as the sample $s_0$ to be encoded.

Table 1: Sender encodes a symbol $s_0$ using Bits Back with ANS.

| BB-ANS stack | Variables | Operation |
|---|---|---|
| Extra information | $s_0$ | |
| $-\log q(y_0 \mid s_0)$ | $s_0, y_0$ | Draw sample $y_0 \sim q(y \mid s_0)$ from the stack. |
| $-\log p(s_0 \mid y_0)$ | $y_0$ | Encode $s_0 \sim p(s \mid y_0)$ onto the stack. |
| $-\log p(y_0)$ | | Encode $y_0 \sim p(y)$ onto the stack. |

This process is clearly invertible, by reversing the order of operation and replacing encodes with decodes and sampling with encoding. Furthermore it can be repeated; the ANS stack at the end of encoding is still an ANS stack, and therefore can be readily used as the extra information for encoding the next symbol. The algorithm is compatible with any model whose prior, likelihood and (approximate) posterior can be encoded and decoded with ANS. A simple Python implementation of both the encoder and decoder of BB-ANS is given in Appendix C.

## 2.5 ISSUES AFFECTING THE EFFICIENCY OF BB-ANS

A number of factors can affect the efficiency of compression with BB-ANS, and mean that in practice, the coding rate will never be exactly equal to the ELBO. For any algorithm based on AC/ANS, the fact that all probabilities have to be approximated at finite precision has some detrimental effect. When encoding a batch of only a small number of i.i.d. samples, with no 'extra information' to communicate, the inefficiency of encoding the first datapoint may be significant. In the worst case, that of a batch with only one datapoint, the message length will be equal to the log joint, $\log p(s_0, y_0)$. Note that optimization of this is equivalent to maximum a posteriori (MAP) estimation. However, for a batch containing more than one image, this effect is amortized. Figure 1 shows an example with 30 samples, where BB-ANS appears to perform well.

Below we discuss two other issues which are specific to BB-ANS. We investigate the magnitude of these effects experimentally in Section 3.2. We find that when compressing the MNIST test set, they do not significantly affect the compression rate, which is typically close to the negative ELBO in our experiments.

### 2.5.1 DISCRETIZING A CONTINUOUS LATENT SPACE

Bits back coding has previously been implemented only for models with discrete latent variables, in Frey (1997). However, many successful latent variable models utilize continuous latents, including the VAE which we use in our experiments. We present here a derivation, based on Mackay (2003), of the surprising fact that continuous latents can be coded with bits back, up to arbitrary precision, without affecting the coding rate. We also briefly discuss our implementation, which as far as we are aware is the first implementation of bits back to support continuous latents. Further discussion can be found in Appendix B.

We can crudely approximate a continuous probability distribution, with density function $p$, with a discrete distribution by partitioning the real line into 'buckets' of equal width $\delta y$. Indexing the buckets with $i \in I$, we assign a probability mass to each bucket of $P(i) \approx p(y_i)\delta y$, where $y_i$ is some point in the $i^{\text{th}}$ bucket (say its centre).

During bits back coding, we discretize both the prior and the approximate posterior using the same set of buckets. We use capital $P$ and $Q$ to denote discrete approximations. Sampling from the discrete approximation $Q(i \,|\, s)$ uses approximately $\log(q(y_i \,|\, s)\delta y)$ bits, and then encoding according to the discrete approximation to the prior $P$ costs approximately $\log(p(y_i)\delta y)$ bits. The expected message length for bits back with a discretized latent is therefore

$$L \approx -\mathbb{E}_{Q(i \,|\, s_0)}\left[\log \frac{p(s_0 \,|\, y_i)p(y_i)\delta y}{q(y_i \,|\, s_0)\delta y}\right]. \tag{3}$$

The $\delta y$ terms cancel, and thus the only cost to discretization results from the discrepancy between our approximation and the true, continuous, distribution. However, if the density functions are sufficiently smooth (as they are in a VAE), then for small enough $\delta y$ the effect of discretization will be negligible.

Note that the number of bits required to generate the latent sample scales with the precision $-\log \delta y$, meaning reasonably small precisions should be preferred in practice. Furthermore, the benefit from increasing latent precision past a certain point is negligible for most machine learning model implementations, since they operate at 32 bit precision. In our experiments we found that increases in performance were negligible past 16 bits per latent dimension.

In our implementation, we divide the latent space into buckets which have equal mass under the prior (as opposed to equal width). This discretization is simple to implement and computationally efficient, and appears empirically to perform well. However, further work is required to establish whether it is optimal in terms of the trade-off between compression rate and computation.

### 2.5.2 THE NEED FOR 'CLEAN' BITS

In our description of bits back coding in Section 2, we noted that the 'extra information' needed to seed bits back should take the form of 'random bits'. More precisely, we need the result of mapping these bits through our decoder to produce a true sample from the distribution $q(y \,|\, s)$. A sufficient

condition for this is that the bits are i.i.d. Bernoulli distributed with probability $\frac{1}{2}$ of being in each of the states $0$ and $1$. We refer to such bits as 'clean'.

During chaining, we effectively use each compressed data point as the seed for the next. Specifically, we use the bits at the top of the ANS stack, which are the result of coding the previous latent $y_0$ according to the prior $p(y)$. Will these bits be clean? The latent $y_0$ is originally generated as a sample from $q(y \mid s_0)$. This distribution is clearly not equal to the prior, except in degenerate cases, so naively we wouldn't expect encoding $y_0$ according to the prior to produce clean bits. However, the true sampling distribution of $y_0$ is in fact the *average* of $q(y \mid s_0)$ over the data distribution. That is, $q(y) \triangleq \int q(y \mid s) p(s) \mathrm{d}s$. This is referred to in Hoffman and Johnson (2016) as the 'average encoding distribution'.

If $q$ is equal to the true posterior, then evidently $q(y) \equiv p(y)$, however in general this is not the case. Hoffman and Johnson (2016) measure the discrepancy empirically using what they call the 'marginal KL divergence' $\mathrm{KL}[q(z)\|p(z)]$, showing that this quantity contributes significantly to the ELBO for three different VAE like models learned on MNIST. This difference implies that the bits at the top the ANS stack after encoding a sample with BB-ANS will not be perfectly clean, which could adversely impact the coding rate.

## 3 EXPERIMENTS

### 3.1 USING A VAE AS THE LATENT VARIABLE MODEL

We demonstrate the BB-ANS coding scheme using a VAE. This model has a multidimensional latent with standard Gaussian prior and diagonal Gaussian approximate posterior:

$$p(y) = N(y; 0, I) \tag{4}$$

$$q(y \mid s) = N(y; \mu(s), \mathrm{diag}(\sigma^2(s))) \tag{5}$$

We choose an output distribution (likelihood) $p(s \mid y)$ suited to the domain of the data we are modelling (see below). The usual VAE training objective is the ELBO, which, as we noted in Section 2.2, is the negative of the expected message length with bits back coding. We can therefore train a VAE as usual and plug it into the BB-ANS framework.

### 3.2 COMPRESSING MNIST

We consider the task of compressing the MNIST dataset (LeCun et al., 1998). We first train a VAE on the training set and then compress the test using BB-ANS with the trained VAE.

The MNIST dataset has pixel values in the range of integers $0, \ldots, 255$. As well as compressing the raw MNIST data, we also present results for stochastically binarized MNIST (Salakhutdinov and Murray, 2008). For both tasks we use VAEs with fully connected generative and recognition networks, with ReLU activations.

For binarized MNIST the generative and recognition networks each have a single deterministic hidden layer of dimension 100, with a stochastic latent of dimension 40. The generative network outputs logits parameterizing a Bernoulli distribution on each pixel. For the full (non-binarized) MNIST dataset each network has one deterministic hidden layer of dimension 200 with a stochastic latent of dimension 50. The output distributions on pixels are modelled by a beta-binomial distribution, which is a two parameter discrete distribution. The generative network outputs the two beta-binomial parameters for each pixel.

Instead of directly sampling the first latents at random, to simplify our implementation we instead initialize the BB-ANS chain with a supply of 'clean' bits. We find that around 400 bits are required for this in our experiments. The precise number of bits required to start the chain depends on the entropy of the discretized approximate posterior (from which we are initially sampling).

We report the achieved compression against a number of benchmarks in Table 2. Despite the relatively small network sizes and simple architectures we have used, the BB-ANS scheme outperforms benchmark compression schemes. While it is encouraging that even a relatively small latent variable model can outperform standard compression techniques when used with BB-ANS, the more

| Dataset | Raw data | VAE test ELBO | BB-ANS | bz2 | gzip | PNG | WebP |
|---|---|---|---|---|---|---|---|
| Binarized MNIST | 1 | 0.19 | **0.19** | 0.25 | 0.33 | 0.78 | 0.44 |
| Full MNIST | 8 | 1.39 | **1.41** | 1.42 | 1.64 | 2.79 | 2.10 |

Table 2: Compression rates on the binarized MNIST and full MNIST test sets, using BB-ANS and other benchmark compression schemes, measured in bits per dimension. We also give the negative ELBO value for each trained VAE on the test set.

important observation to make from Table 2 is that the achieved compression rate is very close to the value of the negative test ELBO seen at the end of VAE training.

In particular, the detrimental effects of finite precision, discretizing the latent (Section 2.5.1) and of less 'clean' bits (Section 2.5.2) do not appear to be significant. Their effects can be seen in Figure 3, accounting for the small discrepancy of around $1\%$ between the negative ELBO and the achieved compression.

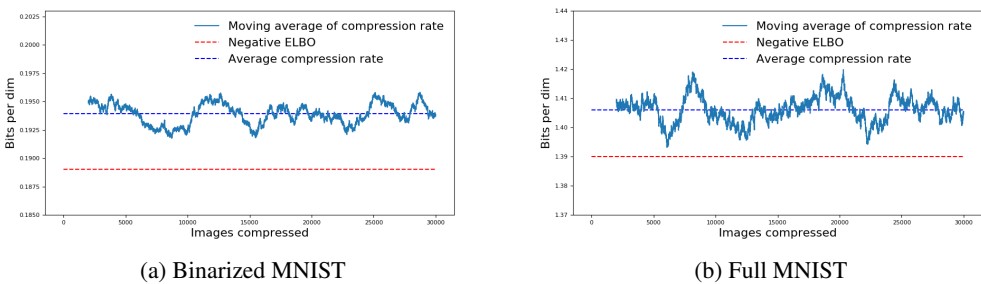

(a) Binarized MNIST            (b) Full MNIST

Figure 3: A 2000 point moving average of the compression rate, in bits per dimension, during the compression process using BB-ANS with a VAE. We compress a concatenation of three shuffled copies of the MNIST test set.

## 4 DISCUSSION

### 4.1 EXTENDING BB-ANS TO STATE-OF-THE-ART LATENT VARIABLE MODELS

Implementing a state-of-the-art latent variable model is not the focus of this work. However, as shown in our experiments, BB-ANS can compress data to sizes very close to the negative ELBO. This means that we can predict the best currently achievable compression using BB-ANS from the reported values of the negative ELBO for state-of-the-art latent variable models. We consider PixelVAE (Gulrajani et al., 2016), a latent variable model with close to state-of-the-art results. We use their reported ELBO on binarized MNIST and the $64 \times 64$ ImageNet dataset introduced in van den Oord et al. (2016).

The predictions are displayed in Table 3, and show that BB-ANS with PixelVAE may have a significantly better compression rate than existing schemes. These predictions are based on the assumption that the discrepancy between compression rate and ELBO will remain small for larger models. We believe this assumption is reasonable, since from the point of view of BB-ANS there are no fundamental differences, apart from dimensionality, between a complex, hierarchical VAE such as PixelVAE and the simple VAEs which we used for our experiments. We leave the experimental verification of these predictions to future work.

Another potential extension of BB-ANS is to time series latent variable models such as hidden Markov models, or latent Gaussian state space models such as those studied in Johnson et al. (2016). Such models could, in principal, be coded with BB-ANS, but the number of 'extra bits' needed in a naive implementation scales with the length of the chain (the total time for a time series model), which could lead to a highly sub-optimal compression rate in practice. It would be useful to have a method for 'interleaving' bits back with the time steps of the model, however it is unclear whether this is possible, and we leave deeper exploration of this problem to future work.

| Dataset | Raw data | BB-ANS with PixelVAE (predicted) | bz2 | gzip | PNG | WebP |
|---|---|---|---|---|---|---|
| Binarized MNIST | 1 | **0.15** | 0.25 | 0.33 | 0.78 | 0.44 |
| ImageNet $64 \times 64$ | 8 | **3.66** | 6.72 | 6.95 | 5.71 | 4.64 |

Table 3: Predicted compression of BB-ANS with PixelVAE against other schemes, measured in bits per dimension.

### 4.2 PARALLELIZATION OF BB-ANS

Modern machine learning models are optimized to exploit batch-parallelism and model-parallelism and run fastest on GPU hardware. Our current implementation of BB-ANS is written in pure Python, is not parallelized and executes entirely on CPU. During encoding/decoding the compression/decompression code is a computational bottleneck, running orders of magnitude slower than the computations of the model probabilities. However, we believe that almost all of the computation in the algorithm could be executed in parallel, on GPU hardware, potentially relieving this bottleneck.

Firstly, our encoder requires computation of the CDF and inverse CDF of the distributions in the model. In the case of a VAE model of binarized MNIST, these are Gaussian and Bernoulli distributions. CDFs and inverse CDFs are already implemented to run on GPU, for many standard distributions, including Gaussian and Bernoulli, in various widely used machine learning toolboxes. Less trivial is the ANS algorithm. However, ANS is known to be amenable to parallelization. Techniques for parallel implementation are discussed in Giesen (2014), and Krajcevski et al. (2016) presents an open source GPU implementation. We leave the performance optimization of BB-ANS, including adapting the algorithm to run on parallel architectures, to future work, but we are optimistic that the marriage of models which are optimized for parallel execution on large datasets with a parallelized and optimized BB-ANS implementation could yield an extremely high performance system.

### 4.3 COMMUNICATING THE MODEL

A neural net based model such as a VAE may have many thousands of parameters. Although not the focus of this work, the cost of communicating and storing a model's parameters may need to be considered when developing a system which uses BB-ANS with a large scale model.

However, we can amortize the one-time cost of communicating the parameters over the size of the data we wish to compress. If a latent variable model could be trained such that it could model a wide class of images well, then BB-ANS could be used in conjunction with such a model to compress a large number of images. This would make the cost of communicating the model weights worthwhile to reap the subsequent gains in compression. Efforts to train latent variable models to be able to model such a wide range of images are currently of significant interest to the machine learning community, for example on expansive datasets such as ImageNet (Deng et al., 2009). We therefore anticipate that this is the most fruitful direction for practical applications of BB-ANS.

We also note that there have been many recent developments in methods to decrease the space required for neural network weights, without hampering performance. For example, methods involving quantizing the weights to low precision (Han et al., 2016; Ullrich et al., 2017), sometimes even down to single bit precision (Hubara et al., 2016), are promising avenues of research that could significantly reduce the cost of communicating and storing model weights.

## 5 CONCLUSION

Probabilistic modelling of data is a highly active research area within machine learning. Given the progress within this area, it is of interest to study the application of probabilistic models to lossless compression. Indeed, if practical lossless compression schemes using these models can be developed then there is the possibility of significant improvement in compression rate over existing methods.

We have shown the existence of a scheme, BB-ANS, which can be used for lossless compression using latent variable models. We demonstrated BB-ANS by compressing the MNIST dataset, achieving compression rates superior to generic algorithms. We have shown how to handle the issue of

latent discretization. Crucially, we were able to compress to sizes very close to the negative ELBO for a large dataset. This is the first time this has been achieved with a latent variable model, and implies that state-of-the-art latent variable models could be used in conjunction with BB-ANS to achieve significantly better lossless compression rates than current methods. Given that all components of BB-ANS are readily parallelizable, we believe that BB-ANS can be implemented to run on GPU hardware, yielding a fast and powerful lossless compression system.

ACKNOWLEDGMENTS

We thank Raza Habib, Harshil Shah and the anonymous reviewers for their feedback. This work was supported by the Alan Turing Institute under the EPSRC grant EP/N510129/1.

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

APPENDIX

## A  BITS BACK CODING

We present here a more detailed derivation of bits back coding.

As before, suppose that a sender and receiver wish to communicate a symbol $s_0$, and they both have access to a generative model with a latent variable, $y$. Suppose both sender and receiver can compute the forward probabilities $p(y)$ and $p(s \mid y)$. How might they communicate a sample $s_0$ from this model?

Naively, the sender may draw a sample $y_0$ from $p(y)$, and encode both $y_0$ and $s_0$ according to the forward model, $p(y)$ and $p(s \mid y_0)$ respectively. This would result in a message length of $-\big(\log p(y_0) + \log p(s_0 \mid y_0)\big)$ bits. The receiver could then decode according to the forward model by first decoding $y_0$ according to $p(y)$ and then decoding $s_0$ according to $p(s \mid y_0)$. However, they can do better, and decrease the encoded message length significantly.

Firstly, if there is some other information which the sender would like to communicate to the receiver, then we may use this to our advantage. We assume the other information takes the form of some random bits. As long as there are sufficiently many bits, the sender can use them to generate a sample $y_0$ by *decoding* some of the bits to generate a sample from $p(y)$, as described in Section 2.1. Generating this sample uses $-\log p(y_0)$ bits. The sender can then encode $y_0$ and $s_0$ with the forward model, and the message length will be $-\big(\log p(y_0) + \log p(s_0 \mid y_0)\big)$ as before. But now the receiver is able to recover the other information, by first decoding $s_0$ and $y_0$, and then encoding $y_0$, reversing the decoding procedure from which the sample $y_0$ was generated, to get the 'bits back'. This means that the net cost of communicating $s_0$, over the other information is $-\log p(s_0 \mid y_0) - \log p(y_0) + \log p(y_0) = -\log p(s_0 \mid y_0)$.

Secondly, note that we can choose any distribution for the sender to sample $y_0$ from, it does not have to be $p(y)$, and it may vary as a function of $s_0$. If we generalize and let $q(\cdot \mid s_0)$ denote the distribution that we use, possibly depending functionally on $s_0$, we can write down the expected message length:

$$L(q) = \mathbb{E}_{q(y \mid s_0)}\big[ -\log p(y) - \log p(s_0 \mid y) + \log q(y \mid s_0)\big] \tag{6}$$

$$= -\mathbb{E}_{q(y \mid s_0)} \log \frac{p(s_0, y)}{q(y \mid s_0)} \tag{7}$$

This quantity is equal to the negative of the evidence lower bound (ELBO), sometimes referred to as the 'free energy' of the model.

Having recognized this equivalence, it is straightforward to show using Gibbs' inequality that the optimal setting of $q$ is the posterior $p(y \mid s_0)$, and that with this setting the message length is

$$L_{\text{opt}} = -\log p(s_0) \tag{8}$$

This is the information content of the sample $s_0$, which by the source coding theorem is the optimal message length. Thus bits back can achieve an optimal compression rate, if sender and receiver have access to the posterior. In the absence of such a posterior (as is usually the case), then an approximate posterior must be used.

We note that Ballé et al. (2018) and Minnen et al. (2018) approach lossless compression with latent variables by generating a latent from an approximate posterior, and encoding according to the prior and likelihood as described above, but not recovering the bits back. Ballé et al. (2018) mention that the cost of coding the hierarchical distribution is only a small fraction of the total coding cost in their setting. This small fraction upper bounds the potential gains from using bits back coding. However, their approach is sub-optimal, even if only slightly, and in the common case where more than one data-point is being encoded they would gain a better compression rate by using BB-ANS.

# B DISCRETIZATION

As we discussed in Section 2.1, the coding scheme we wish to use, ANS, is defined for symbols in a finite alphabet. If we wish to encode a continuous variable we must restrict it to such a finite alphabet. This amounts to discretizing the continuous latent space.

In choosing our discretization, it is important to note the following:

- The discretization must be appropriate for the densities that will use it for coding. For example, imagine we were to discretize such that all but one of our buckets were in areas of very low density, with just one bucket covering the area with almost all of the density. This would result in almost all of the latent variables being coded as the same symbol (corresponding to the one bucket with the majority of the density). Clearly this cannot be an efficient discretization.

- The prior $p(y)$ and the approximate posterior $q(y \mid s)$ must share the same discretization.

- The discretization must be known by the receiver before seeing data, since the first step of decoding is to decode $y_0$ according the prior.

We propose to satisfy these considerations, by using the *maximum entropy discretization* of the prior, $p(y)$, to code our latent variable. This amounts to allocating buckets of equal mass under the prior. We visualize this for a standard Gaussian prior in Figure 4.

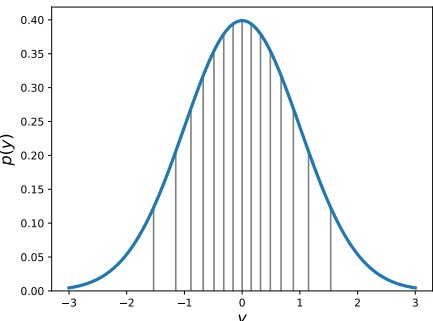

Figure 4: An example of the discretization of the latent space with a standard Gaussian prior, using 16 buckets.

Having the discretization be a function of the prior (which is fixed) allows the receiver to know the discretization up front, which we have noted is necessary. This would not be true for a discretization that depended on the posterior.

This discretization is appropriate for coding according to the prior, since we are maximizing the entropy for this density. However, it is not obvious that it will be appropriate for coding according to the posterior, which it must also be used for.

Note that we can write our the expected message length (negative ELBO) for a single data point as:

$$L(q) = -\mathbb{E}_q(y \mid s_0)\big[\log p(s_0 \mid y)\big] + \mathrm{KL}[q(y \mid s_0)\|p(y)] \tag{9}$$

We can see that minimizing this objective encourages the minimization of the KL divergence between the posterior and the prior. Therefore a trained model will generally have a posterior 'close' (in a sense defined by the KL divergence) to the prior.

This indicates that the maximum entropy discretization of the prior may also be appropriate for coding according to the posterior.

## C  BB-ANS Python Implementation

Figure 5 shows code implementing BB-ANS encoding (as described in Table 1) and decoding in Python. Since the message is stack-like, we use the Pythonic names 'append' and 'pop' for encoding and decoding respectively.

Notice that each line in the decoding 'pop' method precisely inverts an operation in the encoding 'append' method.

The functions to append and pop from the prior, likelihood and posterior could in principle use any LIFO encoding/decoding algorithm. They may, for example, do ANS coding according to a sophisticated autoregressive model, which would be necessary for coding using PixelVAE. The only strict requirement is that each pop function must precisely invert the corresponding append function.

For more detail, including an example implementation with a variational auto-encoder model (VAE), see the repository `https://github.com/bits-back/bits-back`.

```python
def append(message, s):
    # (1) Sample y according to q(y|s)
    #        Decreases message length by -log q(y|s)
    message, y = posterior_pop(s)(message)

    # (2) Encode s according to the likelihood p(s|y)
    #        Increases message length by -log p(s|y)
    message = likelihood_append(y)(message, s)

    # (3) Encode y according to the prior p(y)
    #        Increases message length by -log p(y)
    message = prior_append(message, y)

    return message

def pop(message):
    # (3 inverse) Decode y according to p(y)
    message, y = prior_pop(message)

    # (2 inverse) Decode s according to p(s|y)
    message, s = likelihood_pop(y)(message)

    # (1 inverse) Encode y according to q(y|s)
    message = posterior_append(s)(message, y)

    return message, s
```

Figure 5: Python implementation of BB-ANS encode ('append') and decode ('pop') methods.

