# OpenReview forum: "Practical lossless compression with latent variables using bits back coding"
_ICLR.cc/2019/Conference_

### Official Review · AnonReviewer3 · 2018-11-05
**Very nice paper, but with practical limitations**

**Rating:** 8
**Confidence:** 5

**Review:**

This paper is built on a simple but profound observation: Frey's bits-back coding algorithm can be implemented much more elegantly when replacing arithmetic coding (AC) with asymmetric numerical systems (ANS), a much more recent development not known at the time, simply due to the fact that it encodes symbols in a stack-like fashion rather than queue-like.

This simple observation makes for an elegantly written paper, with promising results on MNIST. I truly enjoyed reading it, and I'm convinced that it will spark some very interesting further work in the field of compression with latent-variable models.

Having said that, I would like to point out some possible limitations of the proposed approach, which I hope the authors will be able to address/clarify:

1. At the beginning of section 2.1, the authors define the symbols as chained conditionals prod_n p(s_n | s_1 ... s_n-1), which is generally permissible in AC as well as ANS, as long as the decoding order is taken into account. That is, in AC, the symbols need to be encoded starting with the first symbol in the chain (s_1), while in ANS, the symbols must be encoded starting with the last symbol in the chain, because the decoding order is inverted.

In their description of BB-ANS, the authors omit the discussion of conditional chains. It is unclear to me if a conditioning of the symbols is feasible in BB-ANS due to the necessity to maintain a strict decoding order. It would be very helpful if the authors could clarify this, and update the paper accordingly, because this could present a serious limitation. For instance, the authors simply extrapolate the performance of their method to PixelVAE; however, this model is autoregressive, so a conditioning of symbols seems necessary. Similarly, in appendix A, the authors mention the work of Minnen et al. (2018), where the same situation would apply, albeit one probabilistic level higher (on encoding/decoding the latents with an autoregressive prior).

2. Furthermore, in both cases (PixelVAE and Minnen et al.), the symbols (s) and latents (y) are defined as jointly conditioned on each other (i.e., computing the posterior on one element of y requires knowledge of all elements of s, and computing the likelihood on one element of s requires knowledge of all elements of y). This seems to imply that all operations pertaining to one data vector (i.e. to one image) would have to be done in a monolithic fashion, i.e.: first sample all elements of y from the stack, then encode all elements of s, and then encode all elements of y. Hence, if the goal is to compress only one image, the algorithm would never get to the point of reusing the "bits back", and the overhead of BB-ANS would be prohibitive. It seems that in the MNIST experiments, the authors avoid this problem by always encoding a large number of images at a time, such that the overhead is amortized.

3. Similarly, although the compression of continuous-valued variables up to arbitrary precision is an exciting development and I do not wish to undermine the importance of this finding, it should be noted that the finer the quantization gets, the larger the potential overhead of the coding scheme will grow. In practice, this would make it necessary to encode more and more images together, in order to still benefit from the method. This would be a good point to make in the discussion.

4. The authors state in the appendix that learned compression methods like Ballé et al. (2018) and Minnen et al. (2018) could be improved by using BB-ANS. The potential gain of BB-ANS for these models seems rather small, though, as the entropy of y must be larger or equal to the entropy of y conditioned on s: H[y] >= H[y|s], the latter of which should represent the potential coding gain. Ballé et al. (2018), however, found that the bits used to encode the hierarchical prior (i.e. H[y]) is only a small fraction of the total bitrate, thus upper bounding the potential gains for this type of model.

Overall, I think this is a well-written, important and elegant paper, and I would like to see it accepted at this conference. If the authors can satisfactorily address some of the above potential limitations, it might turn out to be even better.

---

> ### Author Response · Authors · 2018-11-10
> **Response to reviewer 2, including changes to the paper (Part 1/2)**
>
> Thanks for the extremely well written, thoughtful review. We really appreciate the constructive points made.
>
> We respond to these points below. For ease of reading we've interleaved our responses with the points from the review.
>
> > 1. At the beginning of section 2.1, the authors define the symbols as chained conditionals prod_n p(s_n | s_1 ... s_n-1), which is generally permissible in AC as well as ANS, as long as the decoding order is taken into account. That is, in AC, the symbols need to be encoded starting with the first symbol in the chain (s_1), while in ANS, the symbols must be encoded starting with the last symbol in the chain, because the decoding order is inverted.
>
> > In their description of BB-ANS, the authors omit the discussion of conditional chains. It is unclear to me if a conditioning of the symbols is feasible in BB-ANS due to the necessity to maintain a strict decoding order. It would be very helpful if the authors could clarify this, and update the paper accordingly, because this could present a serious limitation. For instance, the authors simply extrapolate the performance of their method to PixelVAE; however, this model is autoregressive, so a conditioning of symbols seems necessary. Similarly, in appendix A, the authors mention the work of Minnen et al. (2018), where the same situation would apply, albeit one probabilistic level higher (on encoding/decoding the latents with an autoregressive prior).
>
> 1. (RESPONSE) Thanks for raising this. There are (at least) two separate cases to consider regarding encoding of data which is modelled sequentially. One is a latent variable model for i.i.d. data, which contains an autoregressive/sequential prior and/or likelihood and/or posterior distribution. PixelVAE and the model used in Minnen et al. (2018) are both instances of this case.
>
> Another common case is where sequential (e.g. time series) data are modelled with a latent variable model like an HMM, where the latent variable structure is interleaved with and inseparable from the series structure. A sophisticated, VAE-like example of this kind of model is the switching linear dynamical system VAE in [1].
>
> It is straightforward to adapt BB-ANS to deal with the first case. Any model which can be used with ANS can be used as a prior, likelihood or posterior for BB-ANS. As you point out, autoregressive models naturally work for both AC and ANS, and thus PixelVAE and the model used in Minnen et al. (2018) will work fine. You can even nest BB-ANS within BB-ANS (still looking for use cases for this ;-)). One thing that might help to see this is to look at our implementation, which is carefully designed to allow for 'swapping in' component distributions other than the ones we used. See https://github.com/bits-back/bits-back/blob/master/util.py#L152..L166, where the BB-ANS encoder and decoder are explicitly parameterized by the encoders and decoders used for the prior, likelihood and posterior. Each of these encoders and decoders must be based on ANS, and each decoder must exactly invert each encoder, but apart from that there are no additional constraints. An encoder/decoder pair which uses an autoregressive model will work fine (if you're still skeptical, let us know and we can maybe implement a small demo!). Implementing coders for different distributions takes time, and we intend to demonstrate this ability in future work. We have added a sentence clarifying this flexibility to the final paragraph of Section 2.4 of our paper, and an extra (third) paragraph in Appendix C with slightly more detail.
>
> The second case, modelling time series data with an HMM like model, does not appear to be as straightforward. Naively, BB-ANS could be applied treating the whole sequence as monolithic. However we would expect that for a long sequence the number of bits required for generating the latent would be large (it scales with the length of the sequence), and that attaining near optimal compression in this case might not be practically possible. It would be helpful if a method existed to 'interleave' bits back coding with the time series structure of the model. We do have some thoughts in this direction, but we do not feel them conclusive enough to be included in the paper. Nevertheless we have edited the paper, mentioning this limitation of our method in the discussion, in a new paragraph at the end of Section 4.1.

---

> ### Author Response · Authors · 2018-11-10
> **Response to reviewer 2, including changes to the paper (Part 2/2)**
>
> > 2. Furthermore, in both cases (PixelVAE and Minnen et al.), the symbols (s) and latents (y) are defined as jointly conditioned on each other (i.e., computing the posterior on one element of y requires knowledge of all elements of s, and computing the likelihood on one element of s requires knowledge of all elements of y). This seems to imply that all operations pertaining to one data vector (i.e. to one image) would have to be done in a monolithic fashion, i.e.: first sample all elements of y from the stack, then encode all elements of s, and then encode all elements of y. Hence, if the goal is to compress only one image, the algorithm would never get to the point of reusing the "bits back", and the overhead of BB-ANS would be prohibitive. It seems that in the MNIST experiments, the authors avoid this problem by always encoding a large number of images at a time, such that the overhead is amortized.
>
> 2. (RESPONSE) The point you make here is correct. When there is no 'other information' to communicate, and only a single sample, or a very small number of i.i.d. samples are to be compressed, our method is sub-optimal, and we should certainly have highlighted this limitation in our paper. We have extended the first paragraph of Section 2.5 of our paper, emphasizing this point . We have also renamed Section 2.5 to "Issues affecting the efficiency of BB-ANS".
>
> Nevertheless, we believe there are common use cases where a large enough number of (roughly) i.i.d. samples need to be coded, one example being a person's photo library, stored on their computer or smartphone. It is also possible that file meta-data could be used as a source of extra information. We have yet to investigate whether there is sufficient information in typical real world meta-data to resolve this issue.
>
>
>
> > 3. Similarly, although the compression of continuous-valued variables up to arbitrary precision is an exciting development and I do not wish to undermine the importance of this finding, it should be noted that the finer the quantization gets, the larger the potential overhead of the coding scheme will grow. In practice, this would make it necessary to encode more and more images together, in order to still benefit from the method. This would be a good point to make in the discussion.
>
> 3. (RESPONSE) This is a good point, which is definitely worth mentioning. We have added an extra paragraph discussing this point, near the end of Section 2.5.1. We also mention that we found that increasing the precision past around 16 bits yielded no measurable gains in compression rate. We think this is because the models we used (like most machine learning implementations) operated at 32 bit floating point precision.
>
>
>
> > 4. The authors state in the appendix that learned compression methods like Ballé et al. (2018) and Minnen et al. (2018) could be improved by using BB-ANS. The potential gain of BB-ANS for these models seems rather small, though, as the entropy of y must be larger or equal to the entropy of y conditioned on s: H[y] >= H[y|s], the latter of which should represent the potential coding gain. Ballé et al. (2018), however, found that the bits used to encode the hierarchical prior (i.e. H[y]) is only a small fraction of the total bitrate, thus upper bounding the potential gains for this type of model.
>
> 4. (RESPONSE) Thanks a lot for pointing that out. The paragraph in question here is not central to our paper. However we feel that in spite of the limited gains in compression rate from BB-ANS in the case of the two papers mentioned, it's still worth us including this paragraph, particularly because it's quite possible that in future work similar to the two papers mentioned the gain from getting the bits back could be more significant. We've reworded the paragraph, incorporating the bound which you mention.
>
> References
> ---------------
> [1] Johnson, M., Duvenaud, D., Wiltschko, A., Datta, S. and Adams, R. (2016). Composing graphical models with neural networks for structured representations and fast inference. In Advances in Neural Information Processing Systems (NIPS).

---

> > ### Comment · AnonReviewer3 · 2018-11-29
> > **No score change**
> >
> > Thank you for addressing my concerns. Based on this, I am keeping my original score.

---

### Official Review · AnonReviewer1 · 2018-11-05
**Several Questions**

**Rating:** 6
**Confidence:** 3

**Review:**

The paper is very well written and the clarity is overall high. However, I was left with some questions about the significance of this work after reading this paper.

The authors approach the problem in the Bayesian inference framework. Essentially, the message is modeled as a linear neural network with a single latent layer. The authors only specify the distributions for the posterior and prior in the experimental section, where they set them both to Gaussians. This naturally raise the question how is this model different from the probabilistic PCA model? Moreover, I am confused why would it be necessary to introduce an approximation q(y|s) of the posterior p(y|s), when there is a well known closed form expression for Gaussians? Furthermore, this Gaussian model is well known to have non-unique maximum likelihood solution (due to the invariance to an arbitrary orthogonal transformation). How does that influence the addressed compression problem? Going back to equations (1)-(2), if the authors chose different distributions and the need for the ELBO was justified, wouldn’t that lead to an approximate representation? That is, wouldn’t that necessary imply some loss in compression? And if yes, wouldn't then the proposed approach be not a lossless but a lossy compression algorithm? And then why would this particular approach be better than other numerous lossy compression algorithms which the authors cite?

---

> ### Author Response · Authors · 2018-11-07
> **Response to reviewer 1**
>
> Thanks very much for taking the time to read and review our paper.
>
> Summary of our response
> ------------------------------------------
> The first part of the review focuses mainly on modelling questions. We directly address the points made below. However we'd like to emphasize that our paper, and the contributions we make, are not intended to be about modelling, and for more detail on the type of modelling we did, readers should refer to other papers that we cite, such as [1] and [2].
>
> The second part of the review presents an argument which concludes that our method results in lossy compression. This argument is incorrect. Our algorithm is theoretically guaranteed to be lossless, because it comprises a sequence of ANS steps, each of which is lossless. We think the code that we have written, which tests the correctness of decoded data explicitly, provides a strong practical demonstration of this (https://github.com/bits-back/bits-back/blob/master/torch_vae/torch_bin_mnist_compress.py#L89 and https://github.com/bits-back/bits-back/blob/master/torch_vae/torch_mnist_compress.py#L83).
>
> The argument made is highly concerning for us because it suggests that the reviewer has not understood our method, and its significance, at all.
>
> Detailed response
> -----------------------------
> > Essentially, the message is modeled as a linear neural network with a single latent layer.
>
> The message is not modelled. The data (in our experiments, the MNIST dataset), is modelled using a variational auto-encoder (VAE) model. In our experiments we use a generative model with a ReLU non-linearity, as detailed in our paper, Section 3.2, paragraphs 3 and 4. See [1] or [2] for an introduction to these models.
>
> > The authors only specify the distributions for the posterior and prior in the experimental section, where they set them both to Gaussians.
>
> We also specify a likelihood for each of the two different VAE models that we used in our experiments. We do this in Section 3.2, paragraphs 3 and 4. The (approximate) posterior we use is Gaussian but only when conditioned on the observation. The mapping from the observation to the conditional mean and covariance matrix is non-linear.
>
> > This naturally raise the question how is this model different from the probabilistic PCA model?
>
> The key difference is the non-linearity in the conditional distribution (likelihood) p(s | y). We also use discrete distributions for p(s | y), not Gaussian as in PPCA. This is detailed in Section 3.2 of our paper, paragraphs 3 and 4.
>
> > Moreover, I am confused why would it be necessary to introduce an approximation q(y|s) of the posterior p(y|s), when there is a well known closed form expression for Gaussians?
>
> The likelihoods we use contain ReLU non-linearities and there is not conjugacy between the prior and the likelihood, and no known closed form expression for the posterior in this case. Also the observations are not Gaussian.
>
> > Furthermore, this Gaussian model is well known to have non-unique maximum likelihood solution (due to the invariance to an arbitrary orthogonal transformation). How does that influence the addressed compression problem?
>
> Although the review is wrong about the type of model we use, it is true that the model we use also has symmetries which imply that no maximum likelihood solution is unique. As far as we are aware this does not influence the addressed compression problem.
>
> > Going back to equations (1)-(2), if the authors chose different distributions and the need for the ELBO was justified, wouldn’t that lead to an approximate representation? That is, wouldn’t that necessary imply some loss in compression? And if yes, wouldn't then the proposed approach be not a lossless but a lossy compression algorithm?
>
> Any practical model for real world data is approximate. This implies that any lossless compression algorithm using that model will not attain an optimal compression rate. However a sub-optimal compression rate is *not* the same as lossy compression. The 'compression rate' pertains to the length of the message which is output by the algorithm, not the amount of data loss. See [3] Chapters 4-6 for an introduction to these topics.
>
> References
> -----------------
> [1] Kingma, D. P. and Welling, M. (2013). Auto-Encoding Variational Bayes. In Proceedings of the International Conference on Learning Representations (ICLR).
> [2] Rezende, D. J., Mohamed, S., and Wierstra, D. (2014). Stochastic backpropagation and approximate inference in deep generative models. In International Conference on Machine Learning (ICML).
> [3] Mackay, D. (2003). Information Theory, Inference and Learning Algorithms. Cambridge University Press.

---

### Official Review · AnonReviewer2 · 2018-11-13
**Interesting idea, evaluation could be more thorough**

**Rating:** 6
**Confidence:** 4

**Review:**

The main contribution of this paper is to propose an improvement to the bits back (BB) coding scheme by using asymmetric numeral systems (ANS) rather than arithmetic coding for the implementation. ANS is a natural fit with BB since it traverses the coded sequence stack-style rather than FIFO. A second contribution is show how generative models with continuous latent variables can be used (via discretization) within this scheme. The paper is generally well-written, and the explanation in Sec 2.4 was especially clear. However I have some questions about the evaluation and practical application of this scheme.

The comparison in Figure 1 is very compelling, but it would be helpful to have some additional information. In particular, does the size reported for BB-ANS include any overhead related to meta information (e.g., number of images stored, their dimensions, format, etc.)? PNG is a general purpose image file format, so it certainly contains such overhead. This makes it unclear how fair of a comparison we have here. Similarly, bz2 is a general purpose file compression scheme. What file format were the images written as before being compressed? Either of those cases (PNG, bz2) could be opened on any other computer without the need for additional information (just a program that knows how to read/decompress those file formats). On the other hand, the BB-ANS bitstream is not interpretable without the models used when compressing, and as discussed in Sec 4.3, there is certainly additional overhead involved in communicating the model which is not indicated here.

In any case, the compression rate achieved is impressive, but at the same time, not so surprising given that the model was trained on MNIST. Have you checked how well a model trained on a more general image dataset (e.g., ImageNet) compresses other images (e.g., MNIST)?

Sec 3.2 mentions finding that around 400 clean bits are required. How does the performance vary as fewer (or more) clean bits are used? More generally, do you have suggestions for how to determine an appropriate number of clean bits for other scenarios? (E.g., does it depend on the number of images to be compressed? their size? some notion of the entropy of the set of images to be compressed? other factors?)

Also, how does the performance vary with the number of symbols (images) to be compressed? I'd believe that the compression rate approaches the ELBO as the number of compressed images becomes large, but how quickly does this convergence occur? How well does the method do if the VAE is trained using a smaller sample size?

Overall this is an interesting idea, and I believe it could be an excellent lossless compression scheme in scenarios where it's applicable. At the same time, there are many aspects where the paper could be strengthened by providing a more thorough investigation/evaluation.


Minor:
- In Sec 2.1, using p for both a general pdf and the model to be learned (i.e., of both s_n and b_i) is potentially confusing.
- Sec 2.5.1 talks about using uniform quantization (buckets of equal width \delta y), but then Appendix B talks about using (nonuniform) maximum entropy discretization. Which was used in the experiments? In an implementation, the quantization strategy needs to be known by both sender and receiver too, so this is additional meta-information overhead, right?
- The discussion in Sec 4.1 seems very speculative and not particularly convincing.

---

> ### Author Response · Authors · 2018-11-16
> **Response to reviewer 3, including changes to the paper. (Part 3/3)**
>
> > The discussion in Sec 4.1 seems very speculative and not particularly convincing.
>
> We do speculate in this section, but we believe the speculation is well-founded. The purpose of this paper is to demonstrate that we can use latent variable models to perform lossless compression. After demonstrating this, it is sensible (and interesting!) to extrapolate to performance on more advanced latent variable models.  As we point out in 4.1, we can't see any significant difference, apart from the dimensionality of the latent and observed variables, to the BB-ANS scheme of using PixelVAE in place of the smaller VAE we use, so we believe it is a reasonable extrapolation.
>
> We also think it's useful to make falsifiable predictions of this kind, laying down a challenge that might spur further research.

---

> > ### Comment · AnonReviewer2 · 2018-12-19
> > **No score change**
> >
> > Thank you for your responses to my comments.

---

> ### Author Response · Authors · 2018-11-16
> **Response to reviewer 3, including changes to the paper. (Part 2/3)**
>
> > Sec 3.2 mentions finding that around 400 clean bits are required. How does the performance vary as fewer (or more) clean bits are used? More generally, do you have suggestions for how to determine an appropriate number of clean bits for other scenarios? (E.g., does it depend on the number of images to be compressed? their size? some notion of the entropy of the set of images to be compressed? other factors?)
>
> The clean bits are only required to be able to pop a sample from the latent posterior off the BB-ANS stack, and we would expect this requires as many bits as the entropy of this latent posterior, q(y|s). This number depends on the dimensionality of the latent, on the choice of distribution q and on the precision of the latent discretization (we have updated our paper to mention this in the second to last paragraph of Section 2.5.1). It doesn't depend on the number of images to be compressed.
>
> If the number of clean bits used to start the BB-ANS stack is above a certain threshold then the BB-ANS scheme will work, if it is below then it will fail. However, one can avoid the possibility of failure due to insufficient clean bits, and avoid having to work out how many you need in advance, by directly generating the first latents at random (rather than popping them from the stack).
>
> We have updated our paper, rewording paragraph 4 of Section 3.2, to clarify this point.
>
>
> > Also, how does the performance vary with the number of symbols (images) to be compressed? I'd believe that the compression rate approaches the ELBO as the number of compressed images becomes large, but how quickly does this convergence occur?
>
> Figure 3 shows a moving average of the compression rate as more and more images are compressed. By 'compression rate', we mean the change in file size per image added to the chain. It demonstrates that the compression rate does not appear to have an upward or downward trend as more images are compressed. Note that the compression rate is variable image on image because the ELBO itself varies from image to image.
>
> However, because of the clean bits at the start of the chain, the very first image will not be efficiently compressed. In our MNIST experiment this effect is relatively small. The average compression rate, per image, is 0.15 * 784 ~= 150 bits. If we use 400 clean bits at the start, the first image will be compressed (in the worst case) in 400 + 150 = 550 bits. Each subsequent image is compressed in 150 bits on average, so once you've compressed 20 or 30 images the overhead from the first is already fairly small. For thousands of images it will be negligible.
>
> We updated our paper in response to an earlier review which raised this point, discussing this issue in the first paragraph of Section 2.5.
>
>
> > How well does the method do if the VAE is trained using a smaller sample size?
>
> If the VAE is trained on a smaller training set then the model will presumably have worse loss values on the test set, which in turn will produce worse compression performance when using this model with BB-ANS. This is of course relevant to how effective BB-ANS would be in practice, but as before, we see this as a modelling question, not one central to the BB-ANS scheme or this paper.
>
>
> > Sec 2.5.1 talks about using uniform quantization (buckets of equal width \delta y), but then Appendix B talks about using (nonuniform) maximum entropy discretization. Which was used in the experiments?
>
> In the experiments we use the maximum entropy discretization referenced in Appendix B. We also state this in the final paragraph of Section 2.5.1. The equal width discretization was introduced in order to simplify the mathematical derivation. The same (mathematical) argument will also apply to a discretization scheme with unequal bucket widths, since we can map the unequal buckets to equal ones whilst preserving the bucket mass by reparameterizing.
>
>
> > In an implementation, the quantization strategy needs to be known by both sender and receiver too, so this is additional meta-information overhead, right?
>
> Yes, the quantization scheme needs to be known by both the sender and receiver, and this is additional meta-information. However, this will be a neglible amount of information when compared to the BB-ANS scheme as a whole (i.e. being able to run the model and the ANS coding scheme) and the model weights, which also must be known by both the sender and receiver.
>
> If BB-ANS is used as we implement in our experiments, then the quantization is simply dividing the real line into a pre-agreed number of buckets which have equal mass under the prior, which is a standard normal distribution. The quantization operation, including caching to save on recomputation, is implemented in 15 lines of Python here: https://github.com/bits-back/bits-back/blob/master/util.py#L90..L104 .

---

> ### Author Response · Authors · 2018-11-16
> **Response to reviewer 3, including changes to the paper. (Part 1/3)**
>
> Thank you for taking the time to read and review our paper. We address the points raised below. For ease of reading we've interleaved our responses with the points from the review.
>
> > The comparison in Figure 1 is very compelling, but it would be helpful to have some additional information. In particular, does the size reported for BB-ANS include any overhead related to meta information (e.g., number of images stored, their dimensions, format, etc.)? PNG is a general purpose image file format, so it certainly contains such overhead. This makes it unclear how fair of a comparison we have here. Similarly, bz2 is a general purpose file compression scheme. What file format were the images written as before being compressed?
>
> As the review points out, Figure 1 compares our algorithm, which is specialised to MNIST digits, to two general purpose algorithms. The figure, which appears in the introduction, was intended to give the reader an intuitive sense of what the paper is about, rather than provide a rigorous comparison.
>
> For readers seeking more detail, rather than provide more technical details in the paper, which we feel would distract from the flow of the introduction, we've added code to reproduce the figure to the Github repo (https://github.com/bits-back/bits-back/blob/master/make_fig_1.py ) and updated the paper, mentioning the code to reproduce the figure in a footnote on page 2.
>
> To answer the specific points in the review, the BB-ANS bit stream does not include any overhead related to meta information. It was generated using the BB-ANS algorithm detailed in Sec 2.4, which has no provision for meta data. The PNG stream does, as pointed out, contain some meta-data, including (at least) the dimensions of and number of color channels in the images. The bz2 stream contains no metadata and was run directly on the raw bits in the (concatenated) images, with no shape or channel information. We thought this was the fairest way to compare bz2 to BB-ANS.
>
>
>
> > Either of those cases (PNG, bz2) could be opened on any other computer without the need for additional information (just a program that knows how to read/decompress those file formats). On the other hand, the BB-ANS bitstream is not interpretable without the models used when compressing, and as discussed in Sec 4.3, there is certainly additional overhead involved in communicating the model which is not indicated here.
>
> A program that knows how to read/decompress PNG/bz2 formats is analogous to a program that implements BB-ANS and has access to a model. That is to say, a compressed bz2 bitstream is equally as incomprehensible as a BB-ANS bitstream without access to software which knows how the data was compressed. The fact that a person could open bz2 compressed files on any computer now is simply due to the popularity of that compression scheme.
>
> However, it's possible that a program that implements BB-ANS for encoding and contained the model weights could be larger than the analogous program required to encode/decode a format like bz2. We don't rigorously compare these file sizes, because we feel it's outside of the scope of our paper. However we do discuss this issue and potential mitigations in section 4.3.
>
>
> > In any case, the compression rate achieved is impressive, but at the same time, not so surprising given that the model was trained on MNIST. Have you checked how well a model trained on a more general image dataset (e.g., ImageNet) compresses other images (e.g., MNIST)?
>
> This is a very interesting point, and certainly a relevant one if we are to consider using BB-ANS as we would use generic algorithms like PNG. However, we believe that it is essentially a modelling question, and the main focus of this paper is not on modelling questions, but on how to do compression given a model.
>
> To find out how well BB-ANS would compress MNIST data using a model trained on ImageNet, we would not need to run BB-ANS compression with the model. Instead we could just examine the loss values (i.e. ELBO) obtained on MNIST data with a latent variable model trained on ImageNet. That is because we have shown that BB-ANS can losslessly compress data at a rate close to the ELBO value. Therefore this problem is not one related to BB-ANS directly, but instead a modelling question - can we train a model that would have low negative ELBO (loss) values for a wide range of images?

---

### Meta-Review · Area_Chair1 · 2018-12-18
**Novel improved lossless compression scheme using VAEs, with limited empirical validation**

**Confidence:** 3
**Recommendation:** Accept (Poster)

**Metareview:**

The paper proposes a novel  lossless compression scheme that leverages latent-variable models such as VAEs. Its main original contribution is to improve the bits back coding scheme [B. Frey 1997] through the use of asymmetric numeral systems (ANS) instead of arithmetic coding. The developed practical algorithm is also able to use continuous latents. The paper is well written but the reader will benefit from prior familiarity with compression schemes. Resulting message bit-length is shown empirically to be close to ELBO on MNIST. The main weakness pointed out by reviewers is that the empirical evaluation is limited to MNIST and to a simple VAE, while applicability to other models (autoregressive) and data (PixelVAE on ImageNet) is only hinted to and expected bit-length merely extrapolated from previously reported log-likelihood. The work could be much more convincing if its compression was empirically demonstrated on larger and better models and larger scale data. Nevertheless reviewers agreed that it sufficiently advanced the field to warrant acceptance.